# Protective Effects of Early Caffeine Administration in Hyperoxia-Induced Neurotoxicity in the Juvenile Rat

**DOI:** 10.3390/antiox12020295

**Published:** 2023-01-28

**Authors:** Julia Heise, Thomas Schmitz, Christoph Bührer, Stefanie Endesfelder

**Affiliations:** Department of Neonatology, Charité—Universitätsmedizin Berlin, 13353 Berlin, Germany

**Keywords:** hyperoxia, hippocampus, caffeine, postnatal developing brain, newborn rat

## Abstract

High-risk preterm infants are affected by a higher incidence of cognitive developmental deficits due to the unavoidable risk factor of oxygen toxicity. Caffeine is known to have a protective effect in preventing *bronchopulmonary dysplasia* associated with improved neurologic outcomes, although very early initiation of therapy is controversial. In this study, we used newborn rats in an oxygen injury model to test the hypothesis that near-birth caffeine administration modulates neuronal maturation and differentiation in the hippocampus of the developing brain. For this purpose, newborn Wistar rats were exposed to 21% or 80% oxygen on the day of birth for 3 or 5 days and treated with vehicle or caffeine (10 mg/kg/48 h). Postnatal exposure to 80% oxygen resulted in a drastic reduction of associated neuronal mediators for radial glia, mitotic/postmitotic neurons, and impaired cell-cycle regulation, predominantly persistent even after recovery to room air until postnatal day 15. Systemic caffeine administration significantly counteracted the effects of oxygen insult on neuronal maturation in the hippocampus. Interestingly, under normoxia, caffeine inhibited the transcription of neuronal mediators of maturing and mature neurons. The early administration of caffeine modulated hyperoxia-induced decreased neurogenesis in the hippocampus and showed neuroprotective properties in the neonatal rat oxygen toxicity model.

## 1. Introduction

The cognitive outcomes of prematurely born infants are inversely correlated with gestational age and birth weight [1]. A variety of exogenous and endogenous factors may negatively affect cognitive and motor development. Cognitive functions of preterm infants are, on average, lower than those of mature infants [1]. Recent studies underlined that neurological impairments affect almost three-quarters of all life-born children, mainly including speech disorders, intellectual disabilities, attention deficit hyperactivity disorder (ADHD), autism spectrum disorders (ASDs), visual disorders, cerebral palsy, and epilepsy [2,3,4,5,6,7].

The well-characterized adult hippocampal neurogenesis recapitulates the entire process of neuronal development from immature embryonic to mature adult stages. In rodents, some differences, largely cellular, have been found between adult neurogenesis and postnatal neurogenesis (reviewed in [8]): however, recent studies have shown that the neurogenic transcription factors of adult and postnatal neurogenesis are very similar [9]. A major subdivision of the hippocampus is the dentate gyrus (DG), in which neuronal stem cell (NSC) activity and neurogenic processes persist into adulthood. In the subgranular zone (SGZ) of the hippocampus, new neurons are produced which, after functional differentiation, play a central role in learning and memory. Through the definitive control of transcription factors, and subsequently neurotransmitters, radial glia-like neural stem cells self-renew and form lineage-determined neural mitotic intermediate progenitor cells (NPCs). These transform into postmitotic maturing granular neurons and then integrate into the granule cell layer (GCL) as mature granular neurons (reviewed in [10]; for a graphical abstract, see Figure 1). The neurotransmitter gamma-aminobutyric acid (GABA) regulates the further differentiation of the NPCs [11]. The hippocampus, a limbic area involved in learning and memory, is particularly vulnerable to endogenous and exogenous factors, such as oxidative stress [12,13,14], drugs [15], hypoxia [16], and inflammation [17], but also highly sensitive to experience [18]. Impairments during a critical period of brain development can thereby lead to long-term abnormal neural development [19].

Neurodevelopmental deficits and injuries in the immature brain are multi-causal. Oxidative stress nevertheless appears to play a prominent role in this process [14,20], especially since preterm infants are overtly vulnerable to elevated oxygen concentrations, which correlates with neurodevelopmental delay after prematurity [21,22,23]. Already at birth, preterm infants are exposed to oxidative stress as they move from the fetal hypoxic environment to an ex utero condition [24], which hence represents relative hyperoxia to them. Preterm infants are often ventilated in case of respiratory insufficiency, and they have developed neither a mature antioxidant enzyme system [25] nor the capacity to adapt vascular responses to fluctuating oxygen concentrations [26]. Episodes such as mechanical ventilation, supplemental oxygen therapy, infections, and increased inflammation induce oxidative stress [27,28]. Therapeutic strategies for antioxidant prophylaxis to counteract the pathological effects of oxidative stress such as oxidation of biomolecules, inflammation, or cell death, using erythropoietin, vitamins A and E, nitric oxide, N-acetylcysteine, melatonin, and caffeine, did not lead to promising results [29,30,31].

**Figure 1 antioxidants-12-00295-f001:**
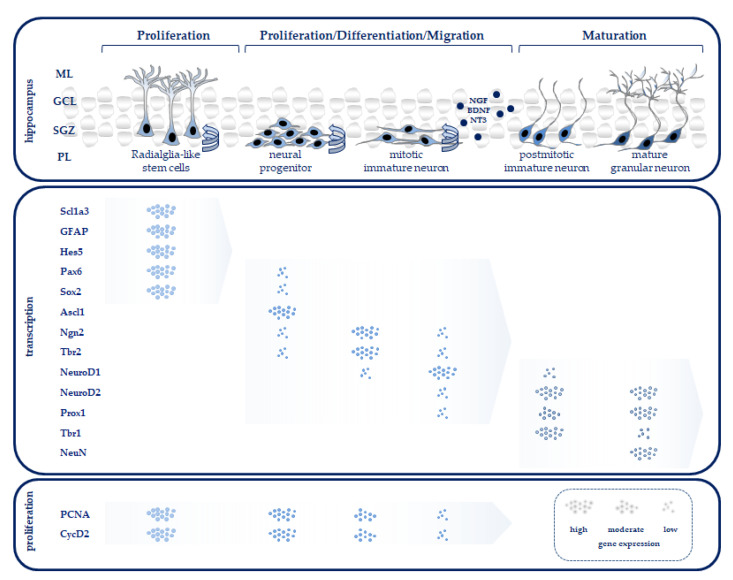
Schematic diagram showing the different stages of neurogenesis in the DG with associated mediators for proliferation, differentiation, and maturation. The thin band between the granule cell layer (GCL) and hilus provides (PL) an environment that allows for the proliferation, specification, and differentiation of the dentate granule cells. Intermediate mitotic progenitor cells arise from asymmetrically dividing radial glia-like stem cells located in the subgranular zone (SGZ) of the DG. After progressive differentiation, dendrite outgrowth of intermediate progenitor cells mature, exit the cell cycle, and differentiate into mature neurons. Most of these regulators play important roles in self-renewal, proliferation, and fate specification during neurogenesis. Transcription factors, as well as neurotrophins, play a central role in neuronal progression during development and coordinate this process via stage-dependent regulation of gene expression, which is dependent on concentration (see the main text for details, modeled after [32]). Abbreviations: Ascl1, achaete-scute family bHLH transcription factor 1; BDNF, brain-derived neurotrophic factor; CycD2, cyclin D2; GFAP, glial fibrillary acidic protein; Hes5, hairy-enhancer-of-split 5; NeuroD1/2, neurogenic differentiation 1/2; NeuN, neuronal nuclear protein; Ngn2, neurogenin 2; NGF, nerve growth factor; NT3, neurotrophin 3; ML, molecular layer; Pax6, paired box 6; PCNA, proliferating cell nuclear antigen; Prox1, Prospero homeobox 1; Sox2, SRY box transcription factor 2; Scl1a3, solute carrier family 1 member 3; SGZ, subgranular zone; Tbr1/2, T-box brain transcription factor 1/2.

In the basal studies by Schmidt et al. [33,34,35] and other research groups [36,37], the standard drug caffeine, which is used to treat apnea in preterm infants, showed a reduction in the incidence of respiratory failure with associated short- and long-term effects, reduction in mortality, reduction in the incidence of bronchopulmonary dysplasia (BPD), and reduction in the duration of mechanical ventilation, possibly reducing ongoing oxidative stress. However, the optimal time point for the initiation of caffeine therapy is still up for debate. Clinical studies indicate that initiation of therapy within the first days of life of preterm infants appears to be most effective [38,39].

We have shown in previous work that caffeine was neuroprotective under hyperoxic insult in 6-day-old rats [12,40]. Notably, we could also show recently that hyperoxia from day 1 of birth massively impaired cerebellar neurogenesis [41]. Caffeine, as a potential antioxidant, was able to rescue cellular neuronal cerebellar damage induced by hyperoxia; however, downstream transcripts important for the migration and differentiation of postmitotic granular cells were irreversibly inhibited by oxidative stress, and this effect was not restored by caffeine [41].

In the current study, we demonstrated that early exposure to high oxygen impaired hippocampal granular cell neurogenesis by analyzing newborn rats with cell-type-specific markers and transcript analysis. Based on this previous studie of cerebellar neurogenesis in the same animals of a near-birth toxic oxygen insult, we hypothesize that early caffeine might save neuronal cells during hippocampal neurogenesis under near-birth toxic oxygen exposure in newborn rats.

## 2. Materials and Methods

### 2.1. Animal Welfare

Time-pregnant Wistar rats were obtained from the Department of Experimental Medicine (FEM, Charité—Universitätsmedizin Berlin, Germany). Adult rats were housed in individual cages under environmentally controlled conditions, with a constant 12 h/12 h light–dark cycle, ambient temperature, and 60% relative humidity, and had ad libitum access to the same food and water. After birth, the neonates were kept with their mothers and fed with breast milk. All animal experimental procedures were evaluated and approved by the local animal welfare authorities (LAGeSo, approval number G-0088/16) and complied with institutional guidelines and ARRIVE guidelines.

The following published studies were generated from identical animal studies and have the same experimental design: [41,42,43].

### 2.2. Oxygen Exposure and Drug Administration

Rat pups from different litters and of both sexes were randomized within 12 h after birth and moved to new cages with the dams. The required sample size was calculated in advance with G*Power V3.1.2 [44]. As previously reported [41,42,43], the further experimental procedures started directly afterwards with exposure under room air (normoxia, NO) or with oxygen-enriched atmosphere (hyperoxia, HY). For a representation of the experimental design, see Appendix A.Oxygen exposure was performed in parallel with 21% or 80% oxygen (OxyCycler BioSpherix, Lacona, NY, USA) from postnatal day (P)0 to P3 (*n* = 6–8) or P0 to P5 (*n* = 6–8) for the newborn rats under further care by the lactating dams. To avoid impairment from oxygen toxicity in lactating dams in order to ensure good care of the rat pups, an exchange between the hyperoxic and normoxic litters occurred every 24 h. Keeping the same exposure times, rats were further divided into four groups: (i) normoxic (NO, control group)—21% oxygen with vehicle (phosphate-buffered saline, PBS); (ii) normoxia with caffeine (NOC)—21% oxygen with caffeine (10 mg/kg, Sigma, Steinheim, Germany); (iii) hyperoxia (HY)—80% oxygen with vehicle (PBS); and (iv) hyperoxia with caffeine (HYC)—80% oxygen with caffeine (10 mg/kg). Applications were intraperitoneally (i.p.) injected as a fixed fraction of their body weight (100 μL/10 g) every 48 h, starting from day of birth (P0). The administration of caffeine or vehicle took place for the pups with a total of three postnatal days of oxygen exposure (P0–P3) on the day of birth (P0) and on P2; and for the rat pups with a total of five days of postnatal oxygen exposure (P0–P5) on the day of birth (P0) and on P2 and P4. Rat pups were examined directly after exposure to oxygen (P3 and P5) or after recovery in room air at P15 (P3_15 and P5_15). No pups died during hyperoxia. Caffeine plasma concentrations and weight profiles were presented in a previous work [42].

### 2.3. Tissue Preparation

At the selected examination time points, which were terminated either immediately after acute oxygen exposure (P3 and P5) or after recovery in room air (P3_15 and P5_15), rat pups were anesthetized with intraperitoneal (i.p.) injection of anesthesia (ketamine (100 mg/kg), xylazine (20 mg/kg), and acepromazine (3 mg/kg)) and transcardially perfused (as previously described [41,42,43]) for further processing before brain removal. After bleeding following stunning, rat pups were decapitated in the cervical region. After opening the skull, the whole brain was removed. To prepare the cerebrum for further molecular or histological analysis, the cerebellum and olfactory brain were removed. For gene-expression studies, the cerebrum tissue was frozen in liquid nitrogen and stored at −80 °C. Perfusion was performed with phosphate-buffered saline (PBS, pH 7.4). For immunohistochemical analysis, perfusion with 4% paraformaldehyde (pH 7.4) was followed by PBS perfusion. After post-fixation at 4 °C for one day, the cerebrum was dehydrated with ethanol and embedded in paraffin to prepare it for histological staining.

### 2.4. RNA Extraction and Quantitative Real-Time PCR

Tissue procurement has already been described [41]. Briefly, for the extraction of RNA, one snap-frozen cerebrum hemisphere per animal was homogenized in peqGOLD RNAPure™ (PEQLAB Biotechnologie, Erlangen, Germany), and the total RNA was isolated per the product protocol. Then 2 µg of RNA was DNase treated and reverse transcribed. Each cDNA sample was diluted 10 times with nuclease-free water and was stored at −20 °C.

PCR was performed with qPCR BIO Mix Hi-ROX (NIPPON Genetics Europe, Düren, Germany). The amplification program was as follows: 50° for 2 min, 94 °C for 2 min, 40 cycles at 94 °C for 5 s, and 62 °C for 25 s. Reactions for each sample were carried out in triplicate in 96-well plates. The detection of PCR products was performed in triplicate in 11 μL reaction mix, each containing 5 μL of qPCR mastermix, 2.5 μL of 1.25 μM of each oligonucleotide primer, 0.5 μL of 5 μM of probe, and 3 μL of cDNA template (17 ng). Probes were labeled with the fluorescent reporter 6-carboxy-fluorescein (6-FAM) at the 5′ end and the fluorescent quencher carboxytetramethylrhodamine (TAMRA) at the 3′ end (BioTez Berlin Buch GmbH, Berlin, Germany). The PCR products of target genes were quantified in real time with the sequences summarized in Table 1. The abundance of each gene was determined relative to the hypoxanthine-guanine phosphoribosyl-transferase (HPRT). The expression of target genes was analyzed with the StepOnePlus real-time PCR system (Applied Biosystems, Carlsbad, CA, USA) according to the 2^−ΔΔCT^ method [45].

### 2.5. Immunohistochemistry

As previously described [41], paraffin-embedded cerebrums were serially sliced to 6 µm thickness and mounted onto SuperFrost Plus coated slides (Menzel, Braunschweig, Germany). Slices were deparaffinized in Roti-Histol (Carl Roth, Karlsruhe, Germany) twice for 10 min each. The slices were subsequently hydrated in ethanol (100%, 100%, 90%, 80%, and 70%) for 3 min each. To demask intracellular epitopes, sections were fixed in citrate buffer (pH 6.0) in a microwave oven for 10 min at 600 W. All slides then cooled at room temperature for half an hour before being washed three times in PBS. The slices were blocked with blocking buffer I (3% goat serum, 0.1% Triton X-100, and 0.05% Twen20 in PBS for NeuN) or blocking buffer II (5% bovine serum albumin and 0.5% Triton X-100 in Tris-buffered saline (TBS) for NeuroD1/PCNA) for 1 h at room temperature. Sections were washed once with PBS or TBS, according to the buffers used for blocking, and subsequently incubated overnight at 4 °C with either monoclonal mouse anti-rat NeuN (1:200, Millipore, MAB377, Darmstadt, Germany) or monoclonal mouse anti-rat NeuroD1 (1:200, Abcam, ab60704, Berlin, Germany) in combination with polyclonal rabbit anti-rat PCNA (1:500, Abcam, ab152112) diluted in antibody diluent (Zymed Laboratories, San Francisco, CA). Slices were washed in PBS or TBS three times, and secondary antibody Alexa Fluor 594-conjugated goat anti-mouse IgG for NeuN staining (Thermo Fisher Scientific, A11032, Dreieich, Germany) or Alexa Fluor 594-conjugated goat anti-rabbi IgG (Thermo Fisher Scientific, A11037), in combination with Alexa Fluor 488-conjugated goat-anti-mouse IgG (Thermo Fisher Scientific, A11029), for NeuroD1/PCNA staining was applied, with consistent 1:200 dilution in antibody diluent (Zymed Laboratories). Sections were incubated for 1 h at room temperature, were consecutively washed once in PBS or TBS, and incubated for 10 min at room temperature after applying 4′,6-diamidino-2-phenylindole (DAPI, Sigma) diluted 1:1000 in PBS or TBS for counterstaining. We mounted (Shandon Immu-Mount, Thermo Fisher Scientific) all sections after three final washes with PBS or TBS. No immunoreactivity was determined in the absence of the primary antibody. Staining per marker was performed on the same day for all experimental groups at each time point. By using serial sections and identifying corresponding section depths, identical areas were analyzed.

The brain slides were analyzed blind, using a Keyence compact fluorescent microscope BZ 9000 with BZ-II Viewer software and BZ-II Analyzer software (Keyence, Osaka, Japan) in the hippocampal dentate gyrus (DG) area, using 10× objective lenses and individual files stitched automatically for each RGB color. Pictures were taken with the same exposure time and contrast/brightness parameters on the same day within one procedure for all experimental groups per study time point. For analysis, two to four separate images of the DG including the hilus per animal were obtained. For quantification of mitotic neurons (NeuroD1/PCNA+), postmitotic neurons (NeuN+), and proliferation capacity (PCNA+), regions of DG with hilus were quantified for each section and were counted manually, using Adobe Photoshop software 22.0.0 (Adobe Systems Software Ireland Limited, Dublin, Republic of Ireland) with minimal previous manipulation of contrast. To determine the region of interest (ROI), the area of the entire hilus, GCL and SGZ of the DG, was marked with an imaginary cut at the beginning of CA3. DAPI was used to visualize the cell nucleus and to mark the granular layer of DG. Mean values per sample were calculated by averaging the values of all sections from the same animal and were used to compare the cell counts of treated animals with those of control animals. The cell numbers counted in the ROI of control animals were used as 100% values, as given in the figure legends.

### 2.6. Statistical Analyses

Box and whisker plots represent the interquartile range (box), with the line representing the median, while whiskers show the data variability outside the upper and lower quartiles. As previously described [32], groups were compared by using one-way analysis of variance (ANOVA), based on a partially non-Gaussian distribution with the Kruskal–Wallis test or based on the assumption that groups do not have equal variances with the Brown–Forsythe test. Depending on which ANOVA test was used, multiple comparisons of means were carried out by using Bonferroni’s, Dunn’s, or Dunnett’s T3 post hoc test. A *p*-value of <0.05 was considered significant. All graphics and statistical analyses were performed by using the GraphPad Prism 8.0 software (GraphPad Software, La Jolla, CA, USA).

## 3. Results

### 3.1. Caffeine Determines the Proliferative Capacity of Near-Birth Hyperoxia-Impaired Intermediate Neurons

In the developing brain, proneural basic helix–loop–helix (bHLH) transcription factors are essential for cell proliferation and equally for neuronal differentiation. Members of the NeuroD family, such as NeuroD1, are key mediators in the regulation of neuronal progenitor cell differentiation and survival [9,46]. To define the mitotic intermediate progenitors in the subgranular zone (SGZ) of the hippocampus, staining for NeuroD1 and PCNA was performed. NeuroD1 is expressed beginning in neuronal progenitors and dynamically with a maximal peak in mitotic immature neurons in co-localization with relevant proliferation markers, here PCNA [47].

High oxygen concentrations both induced inhibition of proliferative capacity per se (Figure 2A,C) and, in particular, significantly inhibited the number of intermediate mitotic neurons in the DG (Figure 2A,B). More specifically, there were drastic decreases of cells capable of division by 29% already after 3 days of exposure, which increased to 66% with an exposure of 5 days. This effect persisted even after recovery to normoxia until P15 (Figure 2C). Considering the NeuroD1/PCNA-positive cells, we see an adequate dramatic decrease in cell numbers in the DG at the acute times P3 and P5 by 31% and by 74%, respectively, persisting until P15 (significant for P3_15, Figure 2B).

Caffeine-treated rats showed significant improvement in cell impairment under hyperoxic exposure. When analyzed at P3 and P5, the percentage of intermediate neurons (Figure 2B), as well as the total proliferative capacity (Figure 2C), was recovered in caffeine-treated animals as compared to vehicle-treated animals. The effect of caffeine faded until P15. Not only did caffeine appear to have lost its effect over long effect times, but caffeine alone also showed unexpected effects. Caffeine-treated rats under normoxia showed a massive impairment of proliferative capacity (decrease to 50% and 41% for P3_15 and P5_15, respectively), as well as counts of neuronal progenitor cells (decrease by 54% for P3_15), comparable to the reduction caused by hyperoxia induction (Figure 2B,C). There were no significant differences in the total number of cells (DAPI-positive cells; see Appendix A). The complete data are shown in full in Appendix A.

Overall, it was shown that hyperoxia resulted in a significant reduction of the neuronal progenitor pool in the DG, as well as the proliferative capacity, in newborn rats of the same level, as measured by the number of NeuroD1-expressing cells, and that the hyperoxia-induced decrease of the progenitor cell pool in the GCL was also directly inhibited by caffeine when applied in parallel, but it decreased in effect size over time until P15.

### 3.2. Caffeine Rescues NeuN-Positive Mature Granule Neurons Damaged by Near-Birth Short-Term Hyperoxia

NeuN, alternatively known as Rbfox3, belongs to the family of RNA-binding Fox proteins (Rbfox) that regulates alternative RNA splicing. NeuN is expressed exclusively in post-mitotic granular neurons [48].

Under hyperoxia, a significant impairment of NeuN-positive mature neurons was observed at P3 with a decrease in number to 77% and P5 with a decrease to 66% (Figure 3A,B). Three days of 80% oxygen exposure resulted in a persisting decrease in the number of NeuN-positive cells at 75% compared with the normoxic controls (Figure 3A,B).

In the hippocampus, caffeine also showed an effect on neurogenic processes under normoxia. Concomitant caffeine application within the first 5 days of life resulted in a 20% greater decrease than the hyperoxic insult and was 50% compared with control animals. Long-lasting effects were also seen at P15, with only a trend after 5 days of concomitant caffeine medication, but significantly decreased by 33% with 3 days of caffeine application (Figure 3B). The complete data are shown in full in Appendix A.

Here, we reported that both the three- and five-day hyperoxia-induced decreased maturation of progenitor cells into mature granular neurons—an effect which persisted after recovery in room air—were improved by caffeine with early concomitant application.

### 3.3. Caffeine Differentially Modulates Neuronal Transcription Factors of Neurogenic Developmental Stages under Toxic and Non-Toxic Oxygen Concentrations

Transcription factors involved in the regulation of neuronal progenitors, neuronal differentiation and maturation, and synaptogenesis are multifaceted and very specifically balanced. These transcription factors play important roles in the regulation of adult, postnatal, and embryonic neurogenesis [8]. The mode of action and the level of expression can differ at different developmental stages, as well as with regard to the proportion of cellular components [49]. The well-characterized neurogenic niches of the developing brain, such as the subgranular zone of the hippocampus, allow for a detailed assessment of the well-orchestrated processes of neurogenesis for the maintenance of neuronal progenitors and the resulting generation of new neurons [50,51]. Thus, endogenous conditions such as oxidative stress or drugs can be evaluated for their effect on the highly vulnerable neurogenic processes of the postnatal rodent brain and be useful for further insights [14,15,40,52,53].

#### 3.3.1. Neuronal Progenitors Benefit from Prolonged Caffeine Administration after Hyperoxic Injury

Neuronal progenitor cells or radial glial neural stem cells in the DG develop into granular neurons and astrocytes [54,55,56,57]. Neuronal transcription factors in the DG of the hippocampus regulate decisions about cell-cycle quiescence and symmetric or asymmetric self-renewal [18]. Transcriptional regulators of asymmetric cell renewal include the neurogenic transcription factors Pax6, Sox2, and Hes5 and the astrogenic transcription factors GFAP and Scl1a3 [58,59].

The expression of *Scl1a3*, *GFAP*, and *Hes5*, as well as *Sox2*, was not affected by near-birth three-day hyperoxia (Figure 4A–C,E). Hyperoxia reduced the transcript levels of *Pax6* after 3 days of exposure (Figure 4D), as it also reduced *Scl1a3*, *Hes5*, *Pax6*, and *Sox2* after 5 days of exposure (Figure 4A,C–E). The reduced mRNA expression persisted until P15 for *Scl1a3* and *Hes5* (Figure 4A,C). Remarkably, the 3-day hyperoxic insult after recovery under normoxia resulted in significant upregulation of *Scl1a3*, *Hes5*, *Pax6*, and *Sox2* (Figure 4A,C–E). Caffeine significantly upregulated the downregulation induced by the toxic oxygen insult, as well as the upregulation in the opposite direction (Figure 4A,C–E). Caffeine without hyperoxic co-insult significantly decreased mRNA expression of *GFAP*, *Pax6*, and *Sox2* primarily at P3 and P5, and for *Sox2* also persistently up to P15 (Figure 4B,D,E). The induction of mRNA transcription by caffeine alone was additionally detected for *Pax6* at P5 (Figure 4D). The complete data are shown in full in a Appendix A.

Here, we reported that transcripts of astrogenic transcription factors tended to be less modulated by oxygen and caffeine, whereas transcripts of neuronal transcription factors experienced more and prolonged impairment by both oxygen and caffeine.

#### 3.3.2. Caffeine Protects Proliferation Capacity and Tends to Protect More the Late Stages of Intermediate Progenitor Cells after Hyperoxic Injury

Transcription factors such as Sox2 and Pax6 are directly associated with certain genes, and in the context of this study, particularly with Ngn2 and NeuroD1, which are also involved in self-renewal but are more focused on further neuronal differentiation [46]. The differentiating intermediate progenitor cells express Tbr2 [60,61], as well as the proneural marker Ascl1 (synonymously Mash1) [46,62], to then start expressing NeuroD1 in the later differentiation process. NeuroD1 represents a crucial neurogenic transcription factor during hippocampal development, marking the transition from mitotic progenitor to neuroblast, and it is thus essential for the further survival and maturation of granular neurons [63].

Acute hyperoxia, immediately after birth, caused decreased mRNA expression of *Ascl1*, *Ngn2*, and *NeuroD1* in the developing rat brain, primarily and significantly after 3 days of exposure (Figure 5A,B,D), significantly after 5 days of exposure for *Ascl1* (Figure 5A), but impressively trending also for Tbr2 and NeuroD1 (Figure 5C,D). Decreased expression persists until P15 or redevelops for *Ascl1* and *Tbr2* (Figure 5A,C). Remarkably, the three-day or five-day hyperoxic insult also resulted in significant upregulation of *Ascl1*, *Ngn2*, and *NeuroD1* here after recovery under normoxia (Figure 5A,B,D). Caffeine significantly enhanced downregulation induced by the toxic oxygen insult, as well as upregulation in the reverse direction (Figure 5A–D), and in particular very impressively for *NeuroD1* at all study time points. Caffeine without hyperoxic co-insult significantly decreased mRNA expression of *Ascl1*, *Ngn2*, and *Tbr2*, as well as *NeuroD1* mainly at P3 or after 3-day and 5-day application at P15 for Acsl1, *Ascl1*, *Ngn2*, and *Tbr2*, as well as *NeuroD1* (Figure 5A–D). The proliferation capacity and self-renewal is particularly high in neural progenitors and intermediate neurons. To complement the proliferation data from immunohistochemical analyses with PCNA, mRNA expression of *CycD2* was determined (Figure 5E). Hyperoxia significantly reduced *CycD2* transcripts at P3, P5, and after 5 days of hyperoxia at P15. At these times, but also without significant damage at P3_15, *CycD2* expression was significantly improved by caffeine co-application. Caffeine under normoxic exposure impaired *CycD2* expression at P5_15 (Figure 5E). The complete data are shown in full in a Appendix A.

We reported that the transcripts of neurogenic transcription factors of intermediate neurons, as well as CycD2 expressed in dividing cells and by neuronal progenitors, are affected by high toxic oxygen, whereas caffeine mediates predominantly more protective effects at this stage of neuronal neurogenic cell differentiation.

#### 3.3.3. Highest Protection for Mature Granular Neurons by Caffeine after Near-Birth Hyperoxic Injury

Simultaneous expression of the bHLH transcription factor NeuroD2 and the homeobox factor Prox1 is highly specific to the early granule neuron lineage [47,64] and is initiated as early as the late developmental stages of the still mitotic intermediate neurons [65]. Tbr1 is highly expressed exclusively in maturing and postmitotic granular neurons [61].

Transcription factor analyses for post-mitotic neurons showed the most pronounced effect of toxic-high oxygen to which the newborn rats were exposed. The mRNA expression of *NeuroD2*, *Prox1*, and *Tbr1* at P3 and P5 was partially reduced to half under hyperoxia (Figure 6A–C), and this effect was only not recovered for *Tbr1* after 5 days of exposure (Figure 6C). Again, compared with control animals, there was increased expression after hyperoxia at P15 for *Prox1* and *Tbr1* (Figure 6B,C). Most impressively, however, the protective effect of caffeine was evident here, as the majority of the co-treatment under hyperoxia resulted in significant improvement in mRNA expression of *NeuroD2*, *Prox1*, and *Tbr1* at P3 and P5 (Figure 6A–C). Increased transcript levels after hyperoxia were suppressed by caffeine (Figure 6A–C). However, even at the final differentiation stage of neurons in the DG, caffeine administration under normoxia mediated transcript-reducing effects for all transcription factors examined, mainly at P3 and at P3_15 and P_15 (Figure 6A–C). The complete data are shown in full in a Appendix A.

We reported that transcripts of neurogenic transcription factors of postmitotic granular neurons are affected by high toxic oxygen, whereas caffeine mediates persistent protective effects up to P15, especially at near-birth exposure times.

#### 3.3.4. Caffeine Is Protective against Acute-Hyperoxia-Induced Downregulation of Neurotrophins

Neurotrophins are crucial for the regulation of neuronal processes in hippocampal neurogenesis and are very efficient in modulating proliferation, differentiation, and maturation, as well as plasticity. They define the hippocampal neurogenic niche [66]. BDNF plays a fundamental role in the hippocampus in synaptic transmission and plasticity [67]. Similarly, NGF and NT3 modulate the regulation of synaptic plasticity [68,69].

Acute sustained hyperoxia for 3 and 5 days after birth significantly reduced neurotrophin transcripts and only immediately after termination of oxygen exposure. Interestingly, an increase in neurotrophins *BDNF*, *NGF*, and *NT3* is also found everywhere after recovery at normoxia (Figure 7A–C). Caffeine co-application abolishes both the decreased expression at P3 and P5 and the increased expression at P15 (Figure 7A–C). Consistent with all results shown so far, caffeine under normoxia also reduced the expression of mRNA from neurotrophins (Figure 7A–C) mainly at P15, independent of the duration of caffeine administration (Figure 7A–C). The complete data are shown in full inAppendix A.

We reported that transcripts of neurotrophic factors are affected by high toxic oxygen, whereas caffeine mediates persistent protective effects, especially at exposure times close to birth.

## 4. Discussion

In this study, we aimed to determine whether early application of caffeine with concomitant near-birth hyperoxic exposure to rat pups would confirm the antioxidant and neuroprotective effects of caffeine on the developing brain, specifically on hippocampal neurogenesis. Furthermore, caffeine proved to inhibit neurogenesis similarly to toxic high oxygen under normoxic environmental conditions.

Apnea of prematurity (AOP) has a high prevalence in neonates with extremely low gestational age [70], and almost all extremely low-birth-weight infants may be exposed to this medication for extended periods of time [71,72]. The efficacy of caffeine in preterm infants has been demonstrated in clinical trials [73,74], with the aim of starting caffeine therapy soon after birth in preterm infants with a low gestational age <32 weeks [75].

Clinical studies showed that long-term neurological outcomes improved when caffeine was used to treat and protect AOP [33,34,35,73,76,77]. How caffeine affects cellular events and the mechanisms that contribute to the protection of neuronal development can be presented in part, but ultimately it is not fully understood. Suggested explanations, which have been confirmed clinically but mainly in experimental animal studies, are direct effects via a nonspecific inhibitor of adenosine receptor subtypes A1 (A1R) and A2a (A2aR) [78], reduction of endoplasmic reticulum stress [79], fewer intermittent hypoxic events through the reduction of AOP [74,80], and reduction of the duration of ventilation, thereby reducing hyperoxic insults and thus also oxidative stress in combination with the antioxidant properties of caffeine per se [12,42,81,82].

Based on the initial study, the CAP trial by Schmidt et al. [33] and relevant studies in recent years (reviewed in [83]), caffeine has been demonstrated to have neuroprotective effects in preterm infants. These effects blur as preterm infants get older [34,35,84], but still suggest that caffeine-treated preterm infants show better performance in fine motor coordination and visual–motor integration [84,85], although with no significant differences in attention and intelligence [35]. Studies on corresponding prematurity-related animal models confirmed this [12,15,40,86,87], but also showed quite contradictory results [41,87,88,89,90,91]. Since all studies are not adequately comparable due to quite different experimental implementations, a general conclusion for an exclusively neuroprotective effect is rather difficult.

The question then arises whether the neuroprotective effects and the few side effects of caffeine on neurobehavioral development could justify a protective administration in all premature infants and, in this context, to timing the initiation without the need for medical intervention. The timing for the initiation of caffeine therapy considered appropriate is not consistent among experts [92,93], and early caffeine therapy cannot be universally recommended; its indication depends on the individual case of the premature infant. However, it is often initiated within the first hours and days of life [38,39,72,94,95,96,97].

Concretely, for this present study, we investigated whether neuronal cell populations involved in hippocampal neurogenesis in the developing rat brain are susceptible to acute hyperoxia of three and five days immediately after birth and whether differences were evident with respect to hyperoxic exposure duration to changes with survival of 12 or 10 days to P15 with and without caffeine treatment. The phase of rapid brain growth, beginning in humans in the last trimester of gestation, with a maximum at term birth, occurs in the rat within the first two weeks of life [98,99]. The prenatally forming DG of the hippocampus exhibits continued postnatal proliferation of granule cells in the hippocampal formation in rodents and occurs somewhat less in humans [100]. In the rat DG, granule cell development peaks between birth and P21 [101], which corresponds to the period in which extremely premature infants are also exposed to high oxygen concentrations, with recovery by the adapted 36 weeks of gestation [98].

The main findings on the effects of hyperoxia with or without caffeine near birth, comparable to extremely preterm infants, were consistent with other studies in the basic message that a high oxygen concentration had a degenerative effect on neurogenesis in the hippocampus [13,102] and that caffeine was neuroprotective during oxidative stress in the developing and adult hippocampus [12,40,103]. However, even in this setting of studying caffeine per se under normoxic conditions, it was impressively shown that caffeine had negative effects under acute administration, as well as after administration with treatment arrest. The characteristics of the effects of caffeine and hyperoxia, as well as their concomitant use, are summarized in Table 2.

Oxidative stress as a pathological mechanism for developmental impairments of the developing brain is well-known [20]. Oxidative stress is endogenously, as well as exogenously, influenced and controlled and regulated by the balance of production and degradation of reactive oxygen species (ROS) in the cell [104]. ROS accumulation indicates cell damage and cell death. Elevated ROS levels also showed links to brain injury and psychiatric disorders, including neurogenic changes [105,106].

Experimental studies link this effect very well and prove the neurogenic inhibitory effects. Proliferative capacity as a criterion for self-renewal of the NSC pool, as well as for the continuation of differentiation to granular neurons, is reduced. Oxygen toxicity is known to directly regulate NSC cell self-renewal, proliferation, and differentiation through the activation of varied oxygen-sensitive signaling [107,108]. The hyperoxic proliferation impairment persisted after the termination of the oxygen insult, suggesting long-lasting impairment and reduced counter-regulation, although cell-specific impairment of early NSCs may be considered here [109], as well as long-lasting changes in the expression of cell-cycle regulatory proteins after hyperoxia [110]. Caffeine can buffer this effect for the total proliferation capacity only during the acute parallel application and increase the proportion of proliferating cells in the DG, but not for the survival period after two weeks. In mitotic intermediate neurons, caffeine can be observed to abolish the reduced differentiation, here also with a protective effect for the later period. Long-term caffeine application promoted adult NSC proliferation in mice in a dose-dependent manner, with moderate doses causing an inhibitory effect on proliferation, whereas supraphysiological doses induced a promoting effect on proliferation [111]. In comparison, in vitro experiments showed impaired proliferation of human hippocampal progenitor cells after acute caffeine exposure [112]. The mechanisms by which caffeine affects proliferation are only partially explained. The antioxidant properties are certainly one aspect [113], as is the mediation of signaling via the adenosine receptors [114], in respect to different expression profiles of adenosine receptors under exogenous insults [115], but the drastic reduction of the transcription factor Sox2 under hyperoxic insult, as well as under normoxia, may provide the link. Sox2 expression correlates with proliferating NSCs [116]. PI3K/Akt signaling is linked to Sox2, and its expression is positively correlated with NSC in adult neurogenesis. The reduction of Sox2 and, thus, reduced NSC proliferation could be related to reduce PI3K/Akt-pathway activation [117]. Neurotrophic factors, such as BDNF, NGF, and NT3, are important factors for the differentiation and maintenance of the neurogenic niche [66,67], that were also downregulated by hyperoxia per se, but also in combination with caffeine in the long term and were not at comparable levels to the controls. BDNF expression peaks in the first weeks after birth in mice, highlighting the importance of its involvement in neurogenic processes [118]. BDNF knockdown mice exhibited significantly fewer proliferating NSCs in the SGZ compared with wild-type mice, and BDNF knockdown appears to be essential for the progression of neurogenesis [119]. Furthermore, BDNF expression impaired by hyperoxia could also contribute to the significantly decreased proliferation of NSCs and NPCs. In addition, adenosine receptor A2a (A2aR) activation increases the expression of BDNF by regulating PI3K/Akt and ERK1/2 signaling in rat primary cortical neurons and, thus, is also related to neuronal proliferation and neuronal cell differentiation [120,121].

In controlling the formation of new functional neurons in the hippocampus, transcription factors play a key role as regulators of the gene expression of relevant neuronal switch sites. Essential overlapping and cascade-regulated mediators drive neurogenesis, whereas the complex linkage with multiple signaling pathways and the inclusion in stage-specific transcriptional complexes is not yet fully understood. The analysis of neurogenic transcription factors revealed multiple impairments and modulations by hyperoxia, as well as by caffeine. Interestingly, these cannot be superimposed congruently with already known findings.

NSCs form precursors of astrocytic and neuronal lineages. Due to their astrocytic imprinting, they express GFAP and Scl1a3 [122,123]. The progression of neurogenesis is dependent on Hes5, a neural-specific factor, as reduced expression slows activation of neurogenic differentiation and exhibits a broad developmental potential, which enables NSCs to serve for a certain time as primary progenitor cells or as NSC pools [124]. Pax6 regulates cell division of NSCs and, provided with similar roles as Hes5, is thought to play an important role in asymmetric division to new NSCs in self-renewal, as well as with intermediate properties [55]. Likewise, Sox2 is able to preserve neuronal differentiation, as well as maintain multipotency [46]. Conditional ablation of Sox2 in the adult hippocampus reduced NSCs and decreased cell proliferation [125]. Sox2, as a transcription activator and member of SOXB1 factors, modulates a variety of genes involved in neuronal differentiation and is thus able not only to drive its own preconditioning for further progression of neuronal differentiation, but also to regulate other proneural genes, such as Ascl1/Mash1 [62], NeuroD1 [64], and neurogenin1 (Ngn1)/Ngn2 [126]. Ngn2 is transiently expressed in multiple NPC types and is downregulated during the final differentiation stages to promote proneural pathways and to regulate subtype specification [126]. Ngn2 is a critical factor that can induce multiple effects, such as inhibition of Pax6, Ascl1/Mash1, or astrocytic and oligodendroglial differentiation, as well as activation of NeuroD1/D2 and, consequently, Tbr2/1 [126,127,128]. With reference to the proneural transcription factor Ascl1/Mash1, which is expressed in mitotic intermediate neurons [129], it is essential for proliferation and further differentiation into mature neurons, interacting closely with neural progenitor cells [130]. The activation of NSC upregulates Ascl1/Mash1, and inactivation completely blocks the exit from progenitor cell quiescence [131]. Tbr2 expression peaks during differentiation in intermediate neurons. Conditional Tbr2 ablation during adult hippocampal neurogenesis promotes NSC proliferation and thus inhibits the progression of differentiation. Hodge et al. [60] hypothesize that Tbr2 counteracts Sox2-mediated arrest by direct inhibition. An important factor for the balance and function of the hippocampal circuit is the neuronal marker NeuN/RbFox3 [132]. NeuN/Rbfox3 is exclusively expressed in postmitotic granular neurons. Results by Lin et al. [133] indicated that neurogenesis in the hippocampus is reduced in Rbfox3-/- mice, and abnormalities of synaptic structure and function occurred. As differentiation progresses, the transcription factor NeuroD1, which begins expression in intermediate neurons, and Prox1, a transcription factor involved in granule cell maturation, are required for hippocampal neurogenesis by facilitating survival and maturation [47,64,134]. This may be supported by the fact that overexpression of NeuroD1 or Prox1 promoted neuronal differentiation of NPCs in vivo, and, in comparison, conditional ablation inhibited the formation of late intermediate neurons [47,134]. In this regard, NeuroD1 appeared to be critical for the survival of maturing granular neurons. Here, Prox1 is not transiently expressed but persists in differentiated mature granular neurons, probably to maintain neuronal identity with pleiotropic properties [47]. Disruption of this sensitive balance of all neurogenic factors involved impairs the progression of neurogenesis.

What are the assumptions in considering these observed effects of very early exposure to oxygen with and without caffeine on the processes of hippocampal neurogenesis?

First, all neurogenic factors, whether cellular, transcriptional, or neurotrophic, are modulated by high oxygen concentrations in terms of hyperoxia. The complex linkages of the involved mediators, as well as the deposited and included diverse signaling pathways, are reflected in these very different expression patterns of acute hyperoxia and/or after survival and recovery with/without caffeine. The astrocytic factors of NCSs appear less susceptible to oxidative stress, as both the proliferative capacity and the progression of neurogenesis are inhibited, manifesting in an overarching delay in the differentiation of any neurogenic stage.

Second, with regard to the hypothesis to be tested that caffeine counteracts this hyperoxia-induced delay in neuronal maturation, it could be demonstrated in hippocampal tissue slices that both proliferation capacity and oxidative-stress-damaged intermediate, as well as mature, neurons were protected by caffeine. The deposited transcription factors benefited from caffeine more in the case of intermediate and mature neurons, similar to the cellular level. Neurotrophic transcripts could support this protective effect. Mainly for NSC transcription factors, no effects—not even enhancing or damaging effects—were shown. The neuroprotective effects of caffeine thus mediated are presumably based on the sum of various cellular mechanisms, due to the broad pharmacological spectrum and adenosine receptor mediation [78,135], as well as the fact that caffeine as a radical scavenger can inhibit ROS production [81]. Experimental rodent models have demonstrated the neuroprotective effects of caffeine. In neonatal hypoxia-exposed mouse pups, caffeine improved myelination and increased the proportion of immature oligodendrocytes [87]. The neuroprotective effect of caffeine under oxidative stress caused by hypoxia was demonstrated by an increase in oligodendroglial differentiation and maturation through modulation of the adenosine receptor subtype A1 in cultured oligodendrocytes [136]. Moreover, an adenosine receptor subtype A2 antagonist protected rat SGZ progenitor cells in DG from decreased proliferation in an oxygen–glucose-deprivation model [137]. We have shown in previous work that caffeine was neuroprotective under hyperoxic insult in 6-day-old rats [12,40]. In our previous study, using the same animals as in this work on hippocampal neurogenesis, the protective effect of caffeine in the hyperoxic-injured cerebellum was limited to neuronal survival but failed to restore important transcript signatures [41].

Third, our results, as well as data from other studies on opposing caffeine effects on neurogenic processes without other exogenous noxious agents, cannot leave the exclusivity of the neuroprotection of caffeine undiscussed. As shown here, caffeine impaired proliferative capacity in the long term and delayed the differentiation process of intermediate and mature neurons. Similar to the study by Giszas et al. [41] for cerebellar neurogenesis, neurogenic transcripts, as well as neurotrophic transcripts, were also found to be reduced in juvenile survival, depending on the application time. Houghton et al. [112] recently reported that caffeine directly affects human hippocampal progenitor cell proliferation and integrity, and this may indicate a mechanism that could influence cognitive outcomes in the human setting. Negative effects of caffeine from animal models likewise focused on neuronal hippocampal proliferation, with chronic application of caffeine decreasing proliferation of neuronal progenitors in rats [111]. Reduced Sox2 transcription could be a possible explanation here, as caffeine is also associated with PI3K/Akt pathways [138].

Data on the effects via the adenosine receptors, mainly concerning A1R and A2aR, the targets for caffeine at nontoxic doses in the brain [139], are scarce and mostly refer to the subventricular zone (SVZ). Differential effects for A1R have been demonstrated. A1R activation promoted NSCs derived from the SVZ to proliferate through mitogen-activated protein kinase (p38 MAPK) and PI3K/Akt signaling pathways [140], whereas, in another study, adenosine was revealed to reduce neuronal differentiation in the SVZ via A1R, correlating with the downregulation of proneuronal genes [141]. Very contrasting data were also shown by studies on A2aR-mediated modification of neurogenic processes of postnatal neurogenesis. Thus, A2aR activation inhibits the proliferation of primary SVZ-genic neurospheres [142], whereas A2aR knockout mice showed reduced proliferation of newborn hippocampal cells in addition to cognitive impairment [143].

All of these facts illustrate that the modulation of adenosine receptors by caffeine, as well as by intracellular released adenosine, has a very variable influence on neurogenic processes. Adenosine signaling in the brain is very complex, with multiple physiological and also pathophysiological effects. Thus, it is hardly surprising that caffeine, as a non-selective adenosine receptor antagonist, shows highly variable effects during postnatal neurogenesis. If we focus on the possible effects of caffeine on neuromodulator effects, adenosine receptor mediation is thought to have multiple functions, such as effects on critical processes of neuronal signaling and derivatively on cognitive and motor functions, as well as modulation of excitatory glutamatergic neurotransmission (reviewed in [144]). Hyperoxygenation led to a decrease in plasma adenosine concentration, as well as A2aR expression in the brain in a concentration-dependent manner in adult rats, with the assumption that lack of A2aR activation at low receptor expression to vasoconstriction represented an adaptive response to reduce tissue oxygenation and oxidative stress by ROS [145]. In contrast, Soontarapornchai et al. [89] showed that long-term-applied caffeine under normoxic insult caused neuroprotection, evidenced by reduction of oxidative stress and hypermyelination, whereas hyperoxia or hypoxia altered or abolished the pharmacodynamic caffeine effects.

As a limitation, we note that the transcriptional data are not exclusively associated with the hippocampus, as we are aware that transcriptional markers are also expressed in other brain regions of the developing brain. However, because the major proliferating niche during the phase of rapid brain growth is the DG [146], and outside the cerebellum and DG, the proliferating new neurons become non-neuronal cells, such as glial cells, fewer differences in the transcription of genes associated with neurogenesis and, specifically, neuronal-lineage-associated cells is to be expected in whole-brain RNA- extract. Further studies should extend our findings towards region-specific transcriptional analyses in the developing brain. It is also notable that the early developmental stages of the brains in the early postnatal rats used in our experimental model is considered comparable to that of the developing human brain in the last trimester of gestation and is hence useful for studies of toxic insults during neurogenic processes [99,147]. Nevertheless, caution should be exercised when extrapolating these results to humans, since, firstly, the pharmacokinetics of caffeine differ between species, and, secondly, some effects of caffeine can differ between rodents and humans [148].

It can be said, in summary, that oxidative stress persistently affected the neurogenic processes of postnatal neurogenesis and caffeine reversed hyperoxia-induced cellular neuronal injury in the hippocampus. However, neuronal transcripts that are important for the maturation and differentiation of mitotic and postmitotic granular cells were persistently inhibited by caffeine. Because balanced proper postnatal cell proliferation and unstressed neuronal function are critical for hippocampus-dependent cognitive function [132,133], contra-induced exogenous insults affecting hippocampal structure and function, such as oxidative stress [14,20,21,22,23], as well as adenosine-receptor-mediated caffeine [112,149], are possible amplifiers of cognitive/motor deficits and a risk factor for social behavior deficits, or they can act as neuroprotectant. Future in vitro studies with a focus on the molecular effects of caffeine on differential cellular mechanisms may serve to further elucidate neurogenic processes, i.e., by using selective adenosine receptor antagonists and agonists, as well as by the selective knockdown of neurogenic targets in both neuronal and non-neuronal cells.

## 5. Conclusions

However, to date, there is no universally accepted standardized protocol on the optimal dosage and timing of caffeine therapy for extremely and very preterm infants. As demonstrated by this study for the effects of caffeine in the hyperoxia injury model on hippocampal neurogenesis, as well as on cerebellar neurogenesis, as shown in the previous study by Giszaz et al. [41], unexpected caffeine effects occurred in the homeostatic brain in addition to the neuroprotective ones. Further studies are warranted for expanding the knowledge of molecular effects for the effects of caffeine on cognitive development in addition to clinical surveys of pulmonary and neurodevelopmental outcomes with caffeine therapy. Determining the effects of caffeine on proliferation and the neurogenic process, as well as the optimal timing for caffeine administration and thus on cognition, will contribute to a better understanding and development of therapeutic interventions.

## Figures and Tables

**Figure 2 antioxidants-12-00295-f002:**
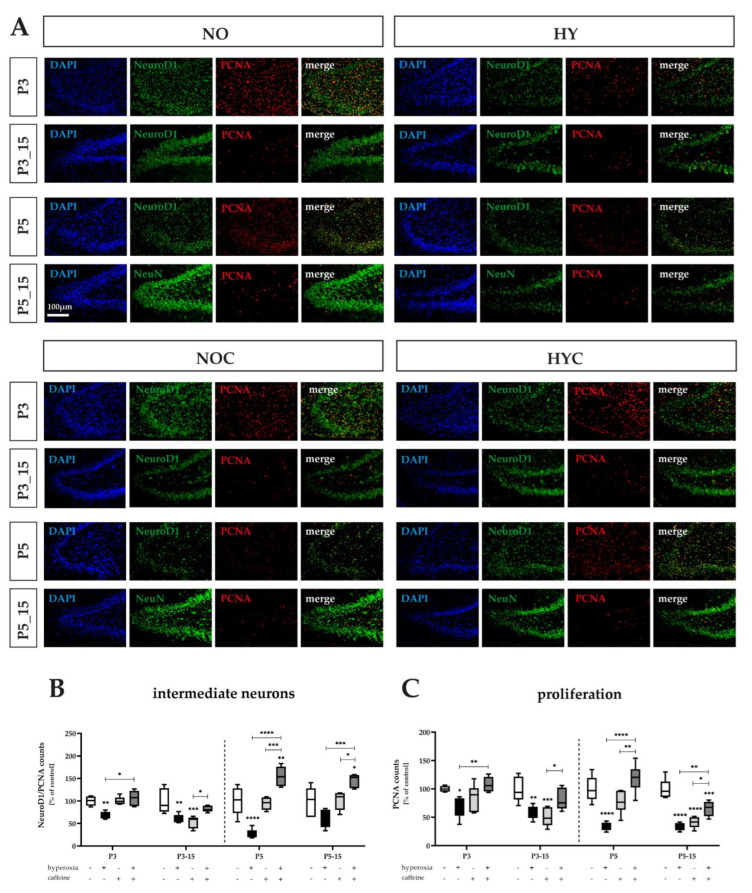
(**A**) Representative hippocampal paraffin sections co−labeled with NeuroD1, PCNA and DAPI of rat pups exposed to normoxia (NO) or hyperoxia (HY) compared to rat pups treated with caffeine (NOC, HYC). Analyses were conducted for 3 days (P3) and 5 days (P5) of postnatal oxygen exposure and after recovery in room air at P15 corresponding to acute exposure after 3 days (P3_15) or 5 days (P5_15) of high oxygen (80%) or normoxia (21%), respectively. Quantification of (**B**) counts of co−labeled NeuroD1− and PCNA− positive intermediate neurons and (**C**) whole proliferating hippocampal cells are presented as hyperoxia exposure (black bars), hyperoxia in combination with caffeine administration (dark gray bars), and caffeine with normoxia (light gray bars) in comparison to normoxia vehicle−treated control animals (white bars). Three (P3) and five days (P5) of lasting hyperoxia affect the proliferation capacity in the entirety of the neuronal cell linages, but here, in this special case, the intermediate neuronal progenitors in the newborn rat DG still impaire until P15 (P3_15 and/or P5_15). Caffeine co−application was able to counteract the impairment of both total proliferative capacity at P3 and P5, and mitotic intermediate neurons additionally up to P5_15. Caffeine under normoxic exposure significantly impaired proliferative capacity in the hippocampus of the 15−day−old rats after both 3−day application (P3_15) and 5−day application (P5_15). Notably, this also affects mitotic intermediate neurons after the 3−day application and recovery period (P3_15). Data are normalized to the level of rat pups exposed to normoxia at each time point (control 100%, white bars), and the 100% values are 71.8 (P3), 84.1 (P3_15), 85.6 (P5), and 52.9 (P5_15) for NeuroD1/PCNA−positive cells, or 157.9 (P3), 407.1 (P3_15), 391.9 (P5), and 249.3 (P5_15) for PCNA−positive cells, respectively. *n* = 5.6/group. * *p* < 0.05, ** *p* < 0.01, *** *p* < 0.001, **** *p* < 0.0001 ((ANOVA, Bonferroni’s post hoc test; Kruskal–Wallis, Dunn’s post hoc test; Brown–Forsythe, Dunnett’s post hoc test).

**Figure 3 antioxidants-12-00295-f003:**
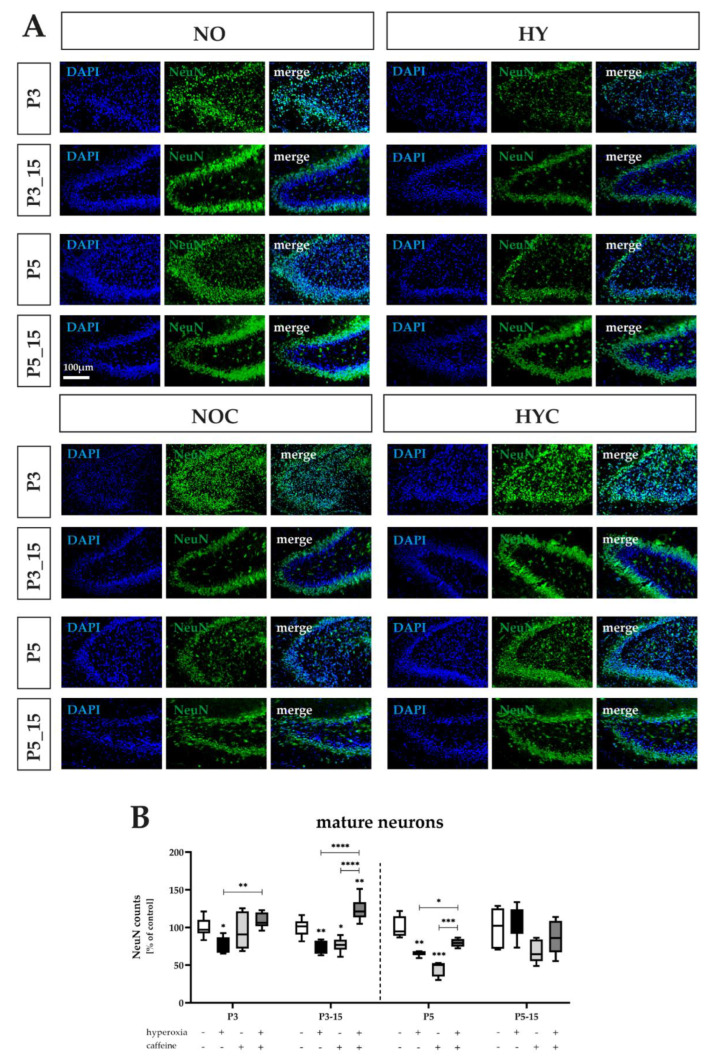
(**A**) Representative hippocampal paraffin sections co−labeled with NeuN and DAPI of rat pups exposed to normoxia (NO) or hyperoxia (HY) compared to rat pups treated with caffeine (NOC, HYC). Analyses were conducted for 3 days (P3) and 5 days (P5) of postnatal oxygen exposure and after recovery in room air at P15 corresponding to acute exposure after 3 days (P3_15) or 5 days (P5_15) of high oxygen (80%) or normoxia (21%), respectively. Quantification of (**B**) counts of labeled NeuN−positive mature neurons are presented as hyperoxia exposure (black bars), hyperoxia in combination with caffeine administration (deep gray bars), and caffeine with normoxia (light gray bars) in comparison to normoxia vehicle−treated control animals (white bars). Three (P3) and five days (P5) lasting hyperoxia affect the proportion of NeuN−positive mature neurons and persists until P15 (only P3_15). Concomitant use of caffeine under hyperoxia was able to counteract the reduction in mature neurons at P3 and P5 and after recovery under normoxia after three days of hyperoxia to P3_15. Caffeine under normoxic exposure significantly impaired neuronal maturation of granular neurons after acute 5−day application (P5), as well as after 3−day application with recovery period (P3_15). Data are normalized to the level of rat pups exposed to normoxia at each time point (control 100%, white bars) and the 100% values are 511.1 (P3), 480.7 (P3_15), 590.1 (P5), and 461.4 (P5_15) for NeuN−positive cells. *n* = 5.6/group. * *p* < 0.05, ** *p* < 0.01, *** *p* < 0.001, **** *p* < 0.0001 (ANOVA, Bonferroni’s post hoc test; Kruskal–Wallis, Dunn’s post hoc test; Brown–Forsythe, Dunnett’s post hoc test).

**Figure 4 antioxidants-12-00295-f004:**
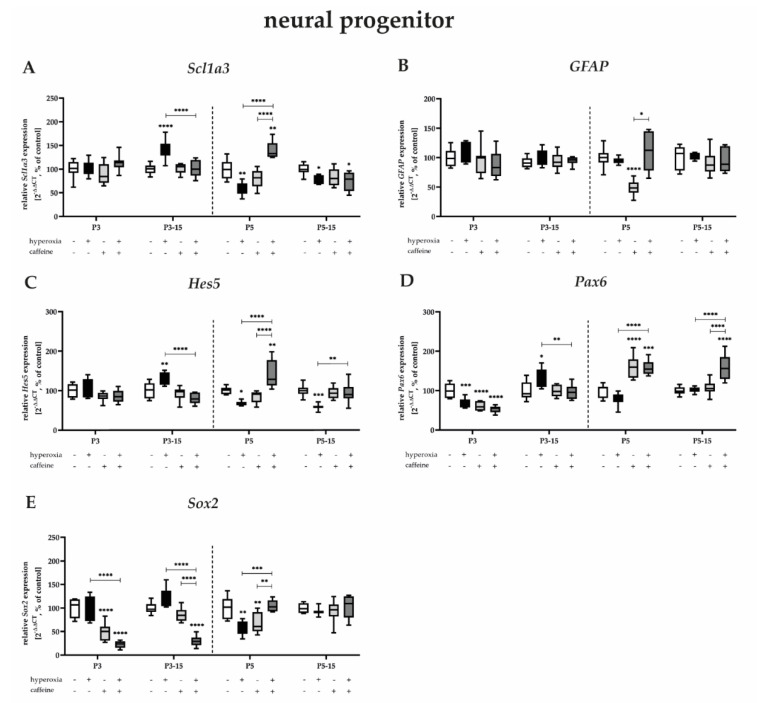
Quantification of cerebral homogenates of one whole hemisphere for mitotic neural progenitor−associated mediators were performed with qPCR for 3 days (P3) and 5 days (P5) of postnatal oxygen exposure and after recovery in room air at P15 corresponding to acute exposure after 3 days (P3_15) or 5 days (P5_15) of high oxygen (80%) or normoxia (21%), respectively. Quantification of gene expressions are presented as hyperoxia exposure (black bars), hyperoxia in combination with caffeine administration (deep gray bars), and caffeine with normoxia (light gray bars) in comparison to normoxia vehicle−treated control animals (white bars). Data are normalized to the level of rat pups exposed to normoxia at each time point (control 100%, white bars). *n* = 7–8/group. * *p* < 0.05, ** *p* < 0.01, *** *p* < 0.001, **** *p* < 0.0001 (ANOVA, Bonferroni’s post hoc test; Kruskal–Wallis, Dunn’s post hoc test; Brown–Forsythe, Dunnett’s post hoc test).

**Figure 5 antioxidants-12-00295-f005:**
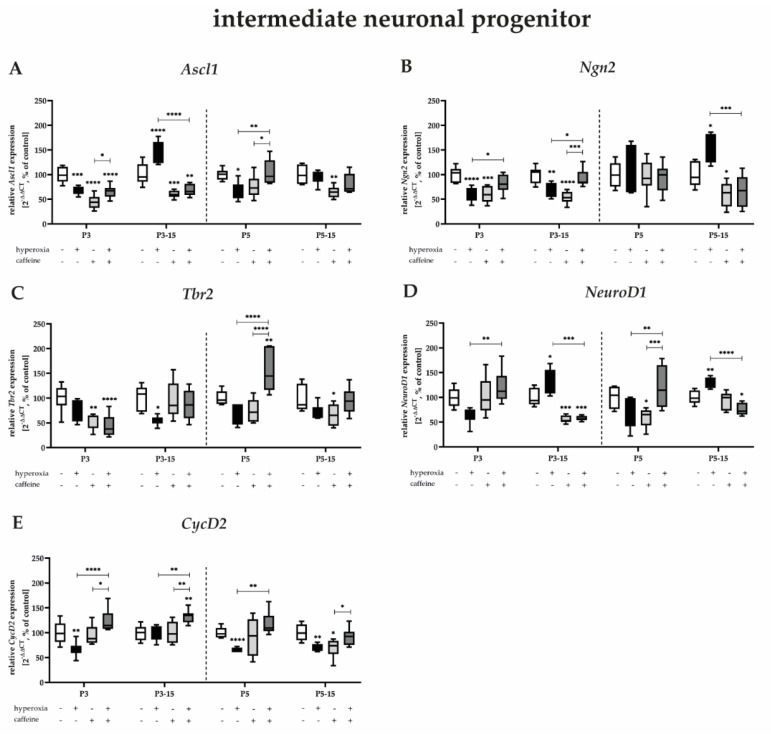
Quantification of cerebral homogenates of one whole hemisphere for intermediate neuronal progenitor−associated mediators were performed with qPCR for 3 days (P3) and 5 days (P5) of postnatal oxygen exposure and after recovery in room air at P15 corresponding to acute exposure after 3 days (P3_15) or 5 days (P5_15) of high oxygen (80%) or normoxia (21%), respectively. Quantification of gene expressions are presented as hyperoxia exposure (black bars), hyperoxia in combination with caffeine administration (deep gray bars), and caffeine with normoxia (light gray bars) in comparison to normoxia vehicle−treated control animals (white bars). Data are normalized to the level of rat pups exposed to normoxia at each time point (control 100%, white bars). *n* = 7–8/group. * *p* < 0.05, ** *p* < 0.01, *** *p* < 0.001, **** *p* < 0.0001 (ANOVA, Bonferroni’s post hoc test; Kruskal–Wallis, Dunn’s post hoc test; Brown–Forsythe, Dunnett’s post hoc test).

**Figure 6 antioxidants-12-00295-f006:**
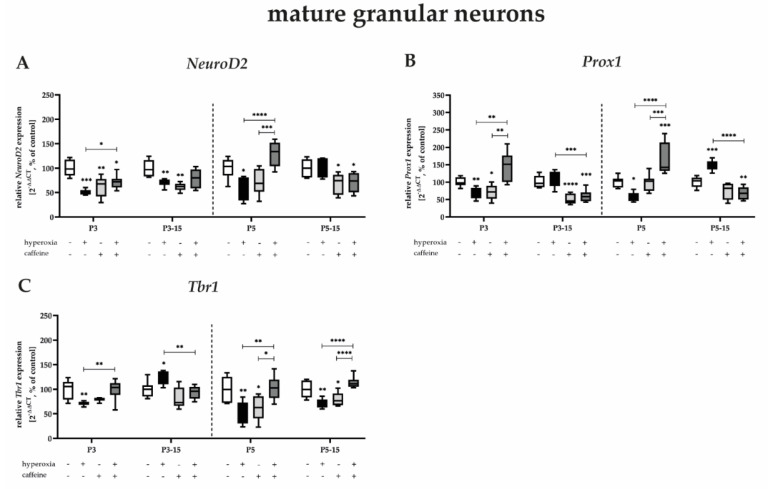
Quantification of cerebral homogenates of one whole hemisphere for mature granular neuron−associated mediators were performed with qPCR for 3 days (P3) and 5 days (P5) of postnatal oxygen exposure and after recovery in room air at P15 corresponding to acute exposure after 3 days (P3_15) or 5 days (P5_15) of high oxygen (80%) or normoxia (21%), respectively. Quantification of gene expressions are presented as hyperoxia exposure (black bars), hyperoxia in combination with caffeine administration (deep gray bars), and caffeine with normoxia (light gray bars) in comparison to normoxia vehicle−treated control animals (white bars). Data are normalized to the level of rat pups exposed to normoxia at each time point (control 100%, white bars). *n* = 7–8/group. * *p* < 0.05, ** *p* < 0.01, *** *p* < 0.001, **** *p* < 0.0001 (ANOVA, Bonferroni’s post hoc test; Kruskal–Wallis, Dunn’s post hoc test; Brown–Forsythe, Dunnett’s post hoc test).

**Figure 7 antioxidants-12-00295-f007:**
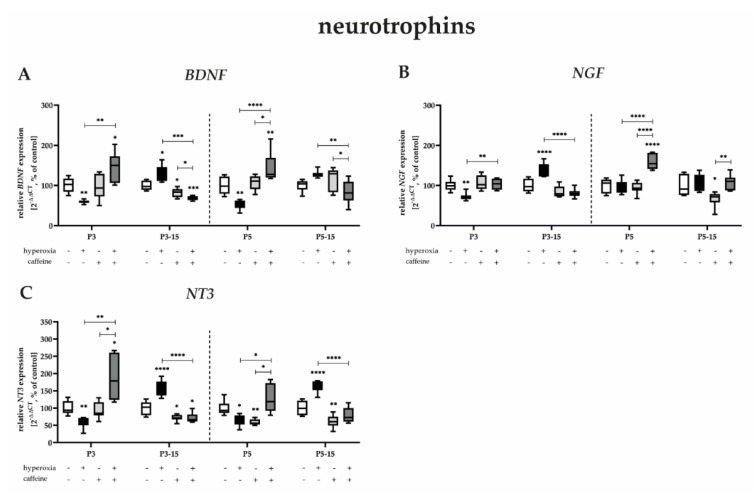
Quantification of cerebral homogenates of one whole hemisphere for neurotrophins were performed with qPCR for 3 days (P3) and 5 days (P5) of postnatal oxygen exposure and after recovery in room air at P15 corresponding to acute exposure after 3 days (P3_15) or 5 days (P5_15) of high oxygen (80%) or normoxia (21%), respectively. Quantification of gene expressions are presented as hyperoxia exposure (black bars), hyperoxia in combination with caffeine administration (deep gray bars), and caffeine with normoxia (light gray bars) in comparison to normoxia vehicle−treated control animals (white bars). Data are normalized to the level of rat pups exposed to normoxia at each time point (control 100%, white bars). *n* = 7–8/group. * *p* < 0.05, ** *p* < 0.01, *** *p* < 0.001, **** *p* < 0.0001 (ANOVA, Bonferroni’s post hoc test; Kruskal–Wallis, Dunn’s post hoc test; Brown–Forsythe, Dunnett’s post hoc test).

**Table 1 antioxidants-12-00295-t001:** Sequences of oligonucleotides.

	Oligonucleotide Sequence 5′-3′	Accession No.
Ascl1 (Mash1)		
Forward	AACTTCAGTGGCTTCGGCTA	NM_022384.1
Reverse	GCCCAGGTTAACCAACTTGA	
Probe	AGCCTTCCACAGCAGCAG	
BDNF		
Forward	TCAGCAGTCAAGTGCCTTTGG	NM_012513.4
Reverse	CGCCGAACCCTCATAGACATG	
Probe	CCTCCTCTGCTCTTTCTGCTGGAGGAATACAA	
CycD2		
Forward	CGTACATGCGCAGGATGGT	NM_199501.1
Reverse	AATTCATGGCCAGAGGAAAGAC	
Probe	TGGATGCTAGAGGTCTGTGA	
GFAP		
Forward	TCTGGACCAGCTTACTACCAACAG	NM_017009.2
Reverse	TGGTTTCATCTTGGAGCTTCTG	
Probe	AGAGGGACAATCTCACACAG	
Hes5		
forward	ATGCTCAGTCCCAAGGAGAA	NM_024383.1
reverse	TAGTCCTGGTGCAGGCTCTT	
probe	CCCAACTCCAAACTGGAGAA	
HPRT		
forward	GGAAAGAACGTCTTGATTGTTGAA	NM_012583.2
reverse	CCAACACTTCGAGAGGTCCTTTT	
probe	CTTTCCTTGGTCAAGCAGTACAGCCCC	
NeuroD1		
forward	TCAGCATCAATGGCAACTTC	NM_019218.2
reverse	AAGATTGATCCGTGGCTTTG	
probe	TTACCATGCACTACCCTGCA	
NeuroD2		
forward	TCTGGTGTCCTACGTGCAGA	NM_019326.1
reverse	CCTGCTCCGTGAGGAAGTTA	
probe	TGCCTGCAGCTGAACTCTC	
NGF		
forward	ACCCAAGCTCACCTCAGTGTCT	NM_001277055.1
reverse	GACATTACGCTATGCACCTCAGAGT	
probe	CAATAAAGGCTTTGCCAAGG	
Ngn2		
forward	AGGCTCAAAGCCAACAACC	XM_008775262.2
reverse	GATGTAATTGTGGGCGAAGC	
probe	CTCACGAAGATCGAGACGCT	
NT3		
forward	AGAACATCACCACGGAGGAAA	NM_031073.3
reverse	GGTCACCCACAGGCTCTCA	
probe	AGAGCATAAGAGTCACCGAG	
Pax6		
forward	TCCCTATCAGCAGCAGTTTCAGT	NM_013001.2
reverse	GTCTGTGCGGCCCAACAT	
probe	CTCCTCCTTTACATCGGGTT	
Prox1		
forward	TGCCTTTTCCAGGAGCAACTAT	NM_001107201.1
reverse	CCGCTGGCTTGGAAACTG	
probe	ACATGAACAAAAACGGTGGC	
Scl1a3 (GLAST)		
forward	CCCTGCCCATCACTTTCAAG	NM_001289942.1
reverse	GCGGTCCCATCCATGTTAA	
probe	CTGGAAGAAAACAATGGTGTGG	
Sox2		
forward	ACAGATGCAGCCGATGCA	NM_001109181.1
reverse	GGTGCCCTGCTGCGAGTA	
probe	CAGTACAACTCCATGACCAG	
Tbr1		
forward	TCCCAATCACTGGAGGTTTCA	NM_001191070.1
reverse	GGATGCATATAGACCCGGTTTC	
probe	AAATGGGTTCCTTGTGGCAA	
Tbr2		
forward	ACGCAGATGATAGTGTTGCAGTCT	XM_006226608.2
reverse	ATTCAAGTCCTCCACACCATCCT	
probe	CACAAATACCAACCTCGACT	

Abbreviations: achaete-scute family bHLH transcription factor 1 (*Ascl1*), brain-derived neurotrophic factor (*BDNF*), cyclin D2 (*CycD2*), glial fibrillary acidic protein (*GFAP*), hairy-enhancer-of-split 5 (*Hes5*), hypoxanthine-guanine phosphoribosyl-transferase (*HPRT*), neurogenic differentiation 1 (*NeuroD1*), neurogenic differentiation 2 (*NeuroD2*), nerve growth factor (*NGF*), neurogenin 2 (*Ngn2*), neurotrophin 3 (*NT3*), paired box 6 (*Pax6*), prospero homeobox 1 (*Prox1*), solute carrier family 1 member 3 (*Scl1a3*), SRY-box transcription factor 2 (*Sox2*), T-box brain transcription factor 1 (*Tbr1*), and T-box brain transcription factor 2 (*Tbr2*).

**Table 2 antioxidants-12-00295-t002:** Summary of the effects of hyperoxia and caffeine on neurogenic processes in the developing brain.

	Effects on Neuronal Cells in the Developing DG
Hyperoxia	Decrease in neural progenitor cell (NPC) pool and proliferative capacity that continued with prolonged hyperoxia in the DG and persisted without regenerative effects. Decreased differentiation of NPCs with sequentially impaired maturation to granular neurons, which persisted without regeneration effects.
Hyperoxia and Caffeine	Decreased proliferation capacity in the DG normalized. Caffeine protected hyperoxia-damaged maturing and mature neurons.
Caffeine	Proliferative capacity is persistently impaired. Caffeine injured maturing and mature neurons.
	**Effects on transcription factors in the developing brain**
Hyperoxia	Impairment of neuronal NSC-related transcription factors was more severe and prolonged than that of astrogenic NSC-related transcription factors. Impairment of neurogenic transcription factors of mitotic-intermediate neurons and postmitotic-mature neurons was more affected by acute oxygen exposure. Induction of transcription factors from neuronal progenitors, intermediate and granular neurons after normoxic survival.
Hyperoxia and Caffeine	NSC-associated transcription factors appear to benefit from prolonged caffeine administration under hyperoxia. Neurogenic transcription factors of maturing NPC and, with higher sensitivity, mature neurons can increase expression levels by caffeine. Caffeine displays no effect or even enhances hyperoxia-downregulated transcription, primarily on earlier neurogenic mediators.
Caffeine	Caffeine reduces transcription factors close to the time of application in NSC-associated transcription factors, whereas NPC- and granular neuronal-associated transcription factors are mostly diminished in expression after long-term survival.
	**Effects on neurotrophic factors in the developing brain**
Hyperoxia	Neurotrophic factors were drastically downregulated by hyperoxia.
Hyperoxia and Caffeine	Caffeine counteracts hyperoxia-repressed transcription of neurotrophic factors.
Caffeine	Neurotrophic factors were drastically downregulated by caffeine.

## Data Availability

The data used to support the findings of this study are available from the corresponding author upon request. The analyzed data used to create the graphs and statistical evaluation are attached in the Appendix A of this work.

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
