# Peer review of "Protective Effects of Early Caffeine Administration in Hyperoxia-Induced Neurotoxicity in the Juvenile Rat"

_antioxidants, 2023, doi:10.3390/antiox12020295_

Round 1

Author Response

Reviewer #1 Report for manuscript #2139477

General impression:

The manuscript „The conflicting role of caffeine supplementation on hyperoxia-induced injury on the cerebral postnatal neurogenesis of newborn rats” by Heise, Bührer and Endesfelder is - as the title implicates - a logical follow-up paper of this research group succeeding the paper by Giszas et al. 2022 (Oxid. Med. Cell Longev.) investigating the conflicting roles of caffeine on hyperoxia-induced injury on cerebellar granular cell neurogenesis in newborn rats.

The paper is quite detailed, frequently accurate and especially refreshingly honest in the “Discussion”, where authors avoid overinterpretation of results and discuss the multifactorial influencing factors, that hamper a simple causal interpretation between drug and outcome.

Despite this positive overall impression, some issues need further clarification and amendments.

Major issues:

Methods:

It is not absolute clear to me, how often caffeine was applied in the individual groups: P3; P5; P3_P15; P5_P15. Especially in the last 2 groups: is the caffeine dose applied every 48h also during normoxic recovery?

Thanks for the advice; we appreciate the opportunity to clarify this in more details. Caffeine administration starts immediately at the day of birth, right before the pups are exposed to hyperoxia or normoxia together with the dam. Pups being exposed until P3 received caffeine or PBS twice, i.e., at P0 and again at P2, while animals exposed until P5 received injections three times, i.e., at P0, P2, and P4. For the recovery phase after the end of hyperoxia until P15, the animals received no further applications. To illustrate the experimental design, I created an overview figure, which I will provide in the supplementary materials (see S-Fig.1). We have added these details to the method section.

Please cite method used to label probes for qPCR

BioTez-Berlin Buch GmbH synthesized the unlabeled as well as the fluorescent dye-labeled oligonucleotides as a service. I have added this information in the method section.

A more detailed description of analysis of IHC is needed. How did authors ensure an unbiased stereological cell count? (see ref: Rieskamp et al. 2022; Neural Regen. Res.17).

Were all slices processed in parallel (at the same day?) to ensure similar staining conditions? Did you control for eventual incomplete antibody penetration?

How was the counting frame area (region of interest, ROI) chosen and standardized?

Thank you for this comment, which gives us the opportunity to present these important aspects in a broader way in the method descriptions. Each embedded cerebrum was as a whole used for serial sections from anterior to posterior. The anterior beginning of the hippocampal region could thus be identified very well for each animal and accordingly matching serial sections were used for the respective staining. The fluorescent levels between the different intervention groups in comparison to the control group did not need to be adjusted, as all animals were immunohistochemically stained in parallel in one experiment and subsequently photographed on the microscope under the same exposure conditions on one day. The area of dentate gyrus to be identified was defined as the total area of the hilus, SGZ and GCL with an imaginary boundary when CA3 begins. This was defined before each analysis using the DAPI-positive cells, drawn in and added as another layer in Photoshop. We did not control for incomplete antibody penetration, as described in Rieskamp et al.. However, we analyzed three serial consecutive tissue sections and thus the corresponding DGs tissue sections. No outliers were detected within the values for each animal or in the standard deviations between animals within a group. Thus, we assume a complete penetration, the even so as we used very thin 6 µm sections instead of 40 µm thickness sections in comparison to the work of Rieskamp et al.. All these aspects are now also included in methods.

Hyperoxia: 3 days and, even more, 5 days of 80% O2 is quite a challenge for newborns. I would anticipate cell death/apoptosis and inflammation. Did authors also stain for cell death markers? Insults (inflammation, hyper-activation, drugs) can also lead to de-maturation of neurons (mature neurons re-express markers of progenitors). Can results be biased by such processes?

This is an interesting and important aspect raised by the reviewer. In any case, this is a big challenge for pups in the phase of rapid brain growth. We as well as others have already shown that both high oxygen concentrations and drugs led to inflammation and cell death during the vulnerable phase of brain development. The present work focused on the neurogenic processes. The analyses of inflammation and cell death are currently part of a continuing analysis.

Based on our data, the injurious effect of hyperoxia and caffeine on the DG is caused by decreased proliferation and by disturbed maturation of neurons. Neuronal loss may thus only be detectable in a subpopulation of neuronal lineage cells and does not necessarily need to be statistically relevant to total cell numbers of the DG.

In our analysis, the DG areas including granular cell layer and hilus represented about 1000 ± 10 % DAPI positive cells.  Given that the count for the neuronal subpopulations labeled with a specific marker, for example NeuroD1/PCNA+ cells, were ranging from 50 to 75 cells within the DG, it can be stated that this numbers lays within the statistical variability of control DG cells, hence making it impossible to find obtain significant differences.

However, this valuable question of the influence of apoptotic and inflammatory processes in terms of neurogenic and reprogramming effects will be considered for further work.

Figure 2: To my opinion the DAPI stains are not comparable in 2A/P3-NO and 2A/P3-HY for instance. Quantification of DAPI+-cells is missing or alternatively, quantification should not only be compared to control animals, but also to the number of DAPI positive cells.

We are grateful for this note. The exposure was not artificially enhanced for the individual representative images, but the DAPI counts are comparable. A count of DAPI+ cells was already performed in parallel with the primary analyses, both to ensure comparability of the cell counts of the investigated markers related to cellular-comparable DGs and to be able to exclude possible cell loss-associated events, although not explicitly investigated as a primary aim of this study. Since no differences were detected, this was rather served as our internal control, which of course seems helpful for open questions. Attached is the DAPI analysis (Fig. S-2), which is provided as supplementary material for publication. A corresponding comment can be found in the script.

Please also indicate statistical significance between hyperoxia-/caffeine- and hyperoxia+/caffeine+ at P5 in the first panel of Figure 2B and use headings for the panels to make it easier for the reader to understand, what part is the difference between the 2 panels in Figure 2B and 2C.

We agree that this is an important aspect. The statistical analyses on NOC to HYC are supplemented accordingly. The labeling of figures 2B and 2C as well as for all figures are now added for better identification. Please note that due to the extension of the multiple comparisons by the analysis from NOC to HYC, the significance level of some investigated parameters has changed. The figures have been revised accordingly and are attached to the resubmission in their revised form.

Please use uniform terms for the group: either always P3_P15 and P5_P15 OR P3_15 and P5_15 (in graphs and text).

The terms were adjusted in the text as well as in the figures and tables.

Lines 286 and following: It is not clear, how the 100% values mentioned in the text were counted.

Cell counts of ROI of control animals were used as 100% values. This is now mentioned in methods.

Figure 4: labeling of axis is too small, please use larger font.

Figure 4 has been revised for better identification and to facilitate direct attribution of results from the manuscript text to the figures (now presented as Figures 4, 5, 6, and 7).

In Figure 4 it is necessary to point out, that whole hemispheres (without olfactory system and cerebellum) were used for RNA isolation, and not only the hippocampus. This fact can also result in different effects. However, it would be technically feasible to use only hippocampi.

Thanks for the comment and pointing out the difference. We are aware of this fact and I would like to discuss this in the following.

It is entirely correct that the data collected cannot be adequately and exclusively associated to the hippocampus and we are aware that the transcriptional markers are also expressed in other brain regions of the developing brain. The following aspects were taken into account in our decision: Neurogenesis as a process strictly orchestrated by transcription factors regulating the progression from NSCs to lineage-committed progenitors and the generation of mature neurons; neurogenesis can be oriented regionally and temporally in the development of the perinatal and postnatal brain (Rice et al., 2000;Hsieh, 2012). The main proliferating niche during the phase of rapid brain growth is the dentate gyrus (Stefovska et al., 2008).Relevant transcription factors also are expressed in other brain regions because they are characteristic for postnatal and adult neurogenesis and corresponding migration processes are underlying. It is, of course, utopian to assume that neuronal as well as other transcription factors have only one single specific role to play. The question of which downstream genes are controlled in complex processes such as migration is not fully understood. For example, NeuroD2 is expressed throughout embryogenesis and into adulthood, even outside of proliferative zones, and functions as a regulator of differentiation and maintenance processes of neurons (Ince-Dunn et al., 2006;Wilke et al., 2012). Other factors, like Tbr2, are strongly associated with differentiation processes from radial glia to immature progenitor cells and further to postmitotic neurons (Hodge et al., 2012).

Unambiguous assignment of altered gene expression via transcriptional analysis would hence not be possible. However, a study by Stefovska et al. (Stefovska et al., 2008) showed in the developing brain (P0 to P15) that the in vivo modulation of GABA receptors changes the proliferation capacity in different brain region, such as cortical sections, thalamus and in the dentate gyrus (subgranular zone and granular cell layer) to the same extent. An impairment after systemic administration of GABA antagonists or agonists might act on different reaction pathways, on different cell types and might affect cells practically in all parts of postnatal brain. Proportion of these cells appears to be small compared to the main proportion of granular cells in the hippocampus. Nevertheless, postnatal cell proliferation is age dependent and most pronounced in the cerebellum and the subgranular zone of the dentate gyrus. Outside of cerebellum and dentate gyrus proliferating new neurons become non-neuronal cells, like glia cells.

Additionally, differences in transcription of neurogenesis-associated genes and specifically neuronal lineage-associated cells are less expected in the whole brain RNA or protein extract (if removal olfactory bulb and cerebellum) as all are affected to the same extent. Significant change of transcript levels may prove the influence of oxidative stress- or pharmacological-induced interruptions in neonatal brain, including possible already migrated cells, in whole hemisphere homogenate.

The reference study Stefovska et al. (Stefovska et al., 2008) for the utility of entire hemispheres for the analysis of gene expression was added to the script. This is now also discussed under the aspect of limitation.

A major issue also seems to be the fact, that rodent neurogenesis in the hippocampus and humans are different. Therefore, this study does not really allow to draw any conclusions for caffeine treatment of human infants with regard to timing, dosage and effect. In addition, pre-term infants will differ in their developmental stage, which makes it even more difficult to find the right timing and dosage of caffeine.

This is an important and certainly essential point of discussion. The transferability from rodents to humans plays a decisive role for many questions, but for basic aspects, despite all limitations, the model we are using is well suited. I will discuss some points in the following.

Animal models in neuroscience research are indispensable and that not only for understanding diseases and developing treatments, but also for obtaining data that cannot be obtained in humans. The importance of a basic understanding of the similarities and differences between brains at different levels of organization is obvious, but it is research on animal models, that will help to translate results from animal data to the human brain.

In general, rodents are more immature at birth in terms of brain development and develop more rapidly after birth than humans, although the basic neural and key developmental processes are very similar (Semple et al., 2013). Of course, this should not be generalized, as humans and rodents differ in the duration of neural development - such as the timing of birth with regards to neural maturation, the relative duration of neurogenesis as well as the differential termination in the maturation of neurotransmitters - and are taken into account in the choice of experimental models. Experimental studies show that the brain development of a 7- to 10-day-old rodent is approximately equivalent to that of an term infant at birth, with postnatal day 1 to 10 in rats corresponding to the last trimester of human gestation. This correlates with the timing of the most rapid brain growth in different species (Dobbing et al., 1979;Semple et al., 2013). In rodents, neurogenesis is still very active in the first two postnatal weeks (Rice et al., 2000). Important developmental processes, such as synaptogenesis, migration as well as myelination, take place in rodents in a very short period after birth and are therefore suitable as a model with a high vulnerability to more profound damage to several developmental processes (Zeiss, 2021).The animal model used in this experimental study is more than suitable for the relevance of the statement on the influence of toxic insults and possible preventive drugs to represent the vulnerable, developing brain of an extremely premature infant. However, we agree with the reviewer that some caution should be exercised in extrapolating these results to humans, firstly because the pharmacokinetics of caffeine differ between species and secondly because the effects of caffeine differ between rodents and humans (Nehlig, 2018). We have also highlighted these issues in the discussion of the paper.

As mentioned before, the “Discussion” points out very well the difficulty to discern cause and effect of interventions (oxidative stress, adenosine effects, phosphordiesterase etc…) in this context. In order to move forward with research in future studies, please provide suggestions, how you would address these difficulties in the future. (For instance, to my mind comes the use of more selective drugs, that isolate individual known effects of caffeine). Many more approaches are possible, please expand on this in the Discussion/Conclusion.

The present study labels neuroprotective caffeine effects under oxygen toxicity insult, whereas under normal physiological conditions caffeine caused inhibitory neurogenic effects. Caffeine has a very broad pharmacological effect, and not only exclusively mediated via non-selective antagonization of adenosine receptors. Caffeine itself modulates both the adenosinergic system and the expression of adenosine receptors. The effect in the brain is primarily mediated by A1 and A2a receptor subtypes, which can also be influenced in their expression and act as heterodimers. In addition, dopaminergic and insulinergic signaling pathways are also modulated. To investigate the molecular effects of caffeine in the homeostatic brain in terms of differential mechanisms acting at the cellular level, analyses with selective antagonists and agonists on a chemical basis would generate new insights but would not do justice to the complexities of caffeine acting pleiotrophically. Because of the divergence between protective and inhibitory properties of caffeine, both in newborn models of the immature organism and in the aging brain, as well as in degenerative diseases such as Alzheimer's disease, the cell-specific molecular mechanisms of both neuronal and non-neuronal cells remain to be uncovered. It is therefore particularly relevant and important to address on a larger scale the effects of caffeine in neuronal versus non-neuronal cells in the immature, homeostatic, and aging brain. Further studies depend on these considerations and our focus will be on complex in vitro experiments with correlated selective antagonists and agonists of adenosine receptors as well as selective knock down.

With respect to the reviewer's commentary, we will mention continuing mechanistic studies, but will avoid a more detailed presentation due to the large amount of information and references.

Minor issues:

Abstract: please do not use abbreviations without explanation in the Abstract (see: BPD)

All abbreviations now appear together with explanations.

Figure Legends: There are several Typos for example: “grey bars”, inconsistencies, grammatical errors or incomplete or -difficult to understand- sentences in the Figure Legends

We reviewed the comments and adjusted them where needed. After consultation with the native-speaking proofreader, no further incomprehensibilities were found in the captions of the illustrations.

All parts: there are several formatting problems (use of different font styles, underlined text: see line 569ff)

The formatting has been corrected.

Numbering: I suggest, that paragraph 3.3. is followed by 3 sub-paragraphs labeled 3.3.1, 3.3.2, and 3.3.3.

We are thankful for the hint. Apparently, some original formatting was changed when the publisher created the template.  The subheadings are now adjusted.

There are several Typos, for instance Line 587: NeuroD1/D1 is meant to be NeuroD1/D2? Please check also the references for complete citations: In Ref.49 f.e. the publication year is missing.

The formatting has been corrected and the references have been checked.

References

Dobbing, J., and Sands, J. (1979). Comparative aspects of the brain growth spurt. Early Hum Dev 3, 79-83.

Hodge, R.D., Nelson, B.R., Kahoud, R.J., Yang, R., Mussar, K.E., Reiner, S.L., and Hevner, R.F. (2012). Tbr2 is essential for hippocampal lineage progression from neural stem cells to intermediate progenitors and neurons. J Neurosci 32, 6275-6287.

Hsieh, J. (2012). Orchestrating transcriptional control of adult neurogenesis. Genes Dev 26, 1010-1021.

Ince-Dunn, G., Hall, B.J., Hu, S.C., Ripley, B., Huganir, R.L., Olson, J.M., Tapscott, S.J., and Ghosh, A. (2006). Regulation of thalamocortical patterning and synaptic maturation by NeuroD2. Neuron 49, 683-695.

Nehlig, A. (2018). Interindividual Differences in Caffeine Metabolism and Factors Driving Caffeine Consumption. Pharmacological Reviews 70, 384-411.

Rice, D., and Barone, S., Jr. (2000). Critical periods of vulnerability for the developing nervous system: evidence from humans and animal models. Environ Health Perspect 108 Suppl 3, 511-533.

Semple, B.D., Blomgren, K., Gimlin, K., Ferriero, D.M., and Noble-Haeusslein, L.J. (2013). Brain development in rodents and humans: Identifying benchmarks of maturation and vulnerability to injury across species. Prog Neurobiol 106-107, 1-16.

Stefovska, V.G., Uckermann, O., Czuczwar, M., Smitka, M., Czuczwar, P., Kis, J., Kaindl, A.M., Turski, L., Turski, W.A., and Ikonomidou, C. (2008). Sedative and anticonvulsant drugs suppress postnatal neurogenesis. Ann Neurol 64, 434-445.

Wilke, S.A., Hall, B.J., Antonios, J.K., Denardo, L.A., Otto, S., Yuan, B., Chen, F., Robbins, E.M., Tiglio, K., Williams, M.E., Qiu, Z., Biederer, T., and Ghosh, A. (2012). NeuroD2 regulates the development of hippocampal mossy fiber synapses. Neural Dev 7, 9.

Zeiss, C.J. (2021). Comparative Milestones in Rodent and Human Postnatal Central Nervous System Development. Toxicologic Pathology 49, 1368-1373.

Reviewer 2 Report

The article titled: „The conflicting role of caffeine supplementation on hyperoxia-induced injury on the cerebral postnatal neurogenesis of new-born rats” by Heise et al. used newborn rats in an oxygen injury model to test the hypothesis that near-birth caffeine administration modulates neuronal maturation and differentiation in the hippocampus of the developing brain. Authors found that systemic caffeine administration significantly counteracted the effects of oxygen insult on neuronal maturation in the hippocampus; however, under normoxia, inhibited the transcription of neuronal mediators of maturing and mature neurons. Manuscript needs improvement:

1. The title itself is misleading and does not refer to the content of the publication. "The conflicting role ...on ...injury" is misleading. Caffeine improves the parameters studied under hyperoxia, but its adverse effects are seen in healthy animals. Please correct the title. I understand that the authors wanted to refer to an earlier publication, but in this case the idea does not work. Please write something like 'A distinct role of...'?

2. please elaborate on the abbreviation "BPD” in the abstract.

3. Materials and methods section - missing figure showing diagram of experiments.

4. Figure 4: The figure is not legible. Please remake it in such a way that it is informative. Maybe show only selected slides and transfer the rest to the supplement? This is just a suggestion, the authors will certainly have their own ideas to improve the readability of the figure.

5. Lines 569-576 - why the lines are underlined - I don't understand?

6. Please elaborate on the conclusions, what I miss here is a reference to the authors' previous studies - which would bundle the results into one whole.

Author Response

Reviewer #2 Report for manuscript #2139477

The article titled: „The conflicting role of caffeine supplementation on hyperoxia-induced injury on the cerebral postnatal neurogenesis of new-born rats” by Heise et al. used newborn rats in an oxygen injury model to test the hypothesis that near-birth caffeine administration modulates neuronal maturation and differentiation in the hippocampus of the developing brain. Authors found that systemic caffeine administration significantly counteracted the effects of oxygen insult on neuronal maturation in the hippocampus; however, under normoxia, inhibited the transcription of neuronal mediators of maturing and mature neurons. Manuscript needs improvement:

The title itself is misleading and does not refer to the content of the publication. "The conflicting role ...on ...injury" is misleading. Caffeine improves the parameters studied under hyperoxia, but its adverse effects are seen in healthy animals. Please correct the title. I understand that the authors wanted to refer to an earlier publication, but in this case the idea does not work. Please write something like 'A distinct role of...'?

It was an idea after evaluating the results, but the objection is understandable and we changed the title to the initial one. We appreciate the constructive comment on this.

Please elaborate on the abbreviation "BPD” in the abstract.

All abbreviations are now explained when appearing the first time, also in the abstract as a seperate section.

Materials and methods section - missing figure showing diagram of experiments.

In accordance with requests from reviewer #1, a corresponding graphic has now been made available as supplementary material and provided with a note in the manuscript.

Figure 4: The figure is not legible. Please remake it in such a way that it is informative. Maybe show only selected slides and transfer the rest to the supplement? This is just a suggestion, the authors will certainly have their own ideas to improve the readability of the figure.

In accordance with reviewer 1's comments, Figure 4 has been revised for better identification (now presented as Figures 4, 5, 6, and 7).

Lines 569-576 - why the lines are underlined - I don't understand?

Apparently, this was affected when formatting was changed by the publisher‘s software to create the template. This has been revised.

Please elaborate on the conclusions, what I miss here is a reference to the authors' previous studies - which would bundle the results into one whole.

We have revised conclusions and added references, accordingly.

Round 2

Reviewer 1 Report

The authors have thoroughly answered individual issues, which in my view needed clarification after the first assessment.

I therefore approve the manuscript in its current form.

Reviewer 2 Report

After corrections, the publication is suitable for printing. I would very much like to ask the authors for one more amendment and that is, please change the wording in the conclusion line 908 "...unexpected neurogenesis-inhibitory effects..." (neurogenesis-inhibitory).